# A sequential model of the contribution of preschool fluid and crystallized cognitive abilities to later school achievement

Philippe Carpentier[1], Geneviève Morneau-Vaillancourt[1], Sophie Aubé[1], Célia Matte-Gagné[1], Anne-Sophie Denault[2], Mara Brendgen[3], Simon Larose[2], Amélie Petitclerc[1], Isabelle Ouellet-Morin[4], René Carbonneau[5], Bei Feng[1], Jean Séguin[6], Sylvana Côté[7], Frank Vitaro[8], Richard E. Tremblay[5], Ginette Dionne[1], Michel Boivin[1]*

1 School of Psychology, Université Laval, Québec, Québec, Canada, 2 Faculty of Education Science, Université Laval, Québec, Québec, Canada, 3 Department of Psychology, Université du Québec à Montréal, Montréal, Québec, Canada, 4 Department of Criminology, Université de Montréal, Montréal, Québec, Canada, 5 Department of Pediatrics and Psychology, Université de Montréal, Montréal, Québec, Canada, 6 Department of Psychiatry, Université de Montréal, Montréal, Québec, Canada, 7 Department of Social and Preventive Medicine, Université de Montréal, Montréal, Québec, Canada, 8 School of Psycho-Education, Université de Montréal, Montréal, Québec, Canada

* michel.boivin@psy.ulaval.ca

**Data Availability Statement:** Data cannot be shared publicly because their are proprietary but may be obtained by filling a request to access from

## Abstract

The present study documented in two distinct population-based samples the contribution of preschool fluid and crystallized cognitive abilities to school achievement in primary school and examined the mediating role of crystallized abilities in this sequence of predictive associations. In both samples, participants were assessed on the same cognitive abilities at 63 months (sample 1, n = 1072), and at 41 and 73 months (sample 2, n = 1583), and then with respect to their school achievement from grade 1 (7 years) to grade 6 (12 years). Preschool crystallized abilities were found to play a key role in predicting school achievement. They contributed substantially to school achievement in the early school years, but more modestly in the later years, due to the strong auto-regression of school achievement. They also mediated the association between fluid abilities and later school achievement in the early grades of school, with the former having modest direct contribution to the latter in the later grades. These findings are discussed regarding their implication for preventive interventions.

## Introduction

School achievement results from a cumulative process of mastering new, as well as improving existing skills [1]. It is a prime factor underpinning a nation's gross domestic product and may be seen as an index of human capital [2]. Generally assessed through standardized tests and teacher assessments of reading, writing or mathematics [3, 4], individual differences in school achievement are established early and tend to persist over time from school entry and onward [5]. School achievement has also been associated with many positive outcomes, including psychosocial adjustment, physical and mental health, perceived happiness, and access to higher education [5–8]. Given the high stability of school achievement and its long-term contribution

the Research Unit on Children's Psychosocial Maladjustment Website (http://www.gripinfo.ca/grip/public/www/etudes/en/dadprocedures.asp) and from the Institut de la Statistique du Québec (https://www.jesuisjeserai.stat.gouv.qc.ca/informations_chercheurs/acces_an.html).

**Funding:** Both studies were supported by grants from the Fonds de recherche du Québec Société et Culture (FRQSC), Fonds de recherche du Québec Santé (FRQS), the Social Sciences and Humanities Research Council of Canada (SSHRC), the Canadian Institutes for Health Research (CIHR). In addition, the QNTS was supported by funding from the National Health Research Development Program, Université Laval, Université de Montréal, Université du Québec à Montréal, and Ste. Justine Hospital's Research Center. The QLSCD was also supported by funding from the Gouvernement du Québec, the Lucie and André Chagnon Foundation, the Robert-Sauvé Research Institute of Health and Security at Work, and the Institut de la Statistique du Québec. P.C. and G.M.-V. were supported by a Doctoral Scholarship from the FRQSC. S.A. is supported by a Joseph-Armand Bombardier Doctoral Scholarship from the SSHRC. M.B., C. M.-G., A.P., and I.O.-M. are supported by the Canada Research Chair Program.

**Competing interests:** The authors have declared that no competing interests exist.

to health, well-being, and economic growth, we need a clear understanding of its early determinants, and of the underlying developmental processes linking them to school achievement.

Both fluid and crystallized cognitive abilities are significant predictors of school achievement [1, 9]. Fluid abilities are derived from basic cognitive processes involved in solving problems that do not require knowledge per se. For example, "fluid" cognitive visual-spatial skills (a central component of non-verbal intelligence), cognitive flexibility (a key component of executive functions), and short-term memory have previously been associated with school achievement [3, 10–15]. Crystallized abilities are knowledge-based competencies, such as letter and number knowledge, presumably acquired through applying fluid abilities in nurturing learning environment [3, 12, 14, 16, 17].

There is a consensus that both fluid and crystallized abilities underlie school achievement. However, their relative contributions, and the timing of these contributions to school achievement over time are still debated [18, 19]. Some have argued that fluid abilities are fundamental predictors of school achievement on the basis that children with learning disorders show deficits in fluid abilities, but not necessarily in crystallized abilities [16]. Preschool visual-spatial cognitive skills and short-term memory at age 5 have been found to predict later numeracy and literacy in school [10]. Considered one central component of fluid cognitive abilities, visual-spatial skills have been shown to predict long-term educational-vocational outcomes in addition to standardized scholastic assessments [20]. Likewise, a meta-analysis revealed that executive functions, including cognitive flexibility, are associated with school achievement in mathematics [11].

On the other hand, early crystallized abilities, such as knowledge of letters, numbers, shapes and colors, have also been shown to predict school achievement in the early school years over and above certain fluid skills such as cognitive visual-spatial skill, cognitive flexibility and short-term memory [14, 17]. A meta-analysis of six longitudinal studies found that early crystallized skills, such as early numeracy and literacy, were the best predictors of school achievement when compared to emotional difficulties, behavior problems, and attentional abilities [1].

Yet, recent findings show that the contributions of fluid and crystallized abilities to school achievement are complex and tend to vary over time. For example, one study showed that the association between IQ tests, including both fluid and crystallized abilities, and achievement in mathematics was generally stable from grade 1 to grade 8, whereas the association between executive function and school achievement grew in force across the elementary school grades [19]. The contribution of crystallized abilities to achievement in mathematics also increased over the same period. However, this study only considered a subset of mathematical crystallized abilities (basic number and arithmetical competencies, knowledge of fractions) in primary school, which precludes any conclusion about the possible contribution of a larger range of preschool crystallized abilities to later school achievement. It is possible that the contribution of preschool crystallized abilities to later school achievement could be limited in time, while the contribution of preschool fluid abilities persists or even increases over time. Indeed, the increased academic requirements associated with schooling may progressively entail more complex reasoning strategies while relying less on early knowledge. In support of this view, a study found that the association between visual-spatial cognitive skills and school achievement was stronger in grades 6 to 9 than in grades 1 to 5 [21].

However, simply comparing the unique contributions of fluid and crystallized skills to school achievement is not the most informative way to understand the developmental processes involved. Fluid abilities, such as executive functions, short-term memory and cognitive visual-spatial skills, have been associated with crystallized abilities, such as numeracy and literacy [22, 23]. Given the nature of these abilities, fluid skills likely contribute to the acquisition

of crystallized abilities [23, 24], which means that the latter could play a mediating role in the contribution of early fluid abilities to later school achievement. This possible sequence, whereby fluid abilities facilitate the acquisition of crystallized abilities, relates to actual views regarding the structure and development of cognitive abilities. Some have indeed suggested that the well-documented interrelation generally found among human cognitive abilities results from mutual interactions between various cognitive processes, starting with the more basic perceptual, memory, and reasoning processes [25, 26]. Skill building theory [27] also posits that simpler skills lay the ground for the later development of more advanced and sophisticated skills or behavior [28, 29]. Accordingly, the dynamic association between fluid and crystallized abilities could be seen as a special case of asymmetrical mutualism [26], whereby crystallized abilities stem from the interaction (or investment) of fluid intelligence and experience, ultimately leading to achievement in school.

There is indeed empirical support for this mediation. For instance, one study showed that numerical competence (crystallized abilities) in kindergarten partly mediated the association between fluid skills (working memory, and processing speed) and achievement in mathematics in grade 1 [30]. Another study found that preschool arithmetic skills partly mediated the association between early fluid abilities (executive function, non-verbal intelligence) and grade 1 mathematics achievement [31]. Finally, one study showed that crystallized abilities, such as literacy and early arithmetic skills accounted for the predictive association between early fluid abilities (IQ and executive functions at ages 3 and 5 years) and achievement in kindergarten [32]. Those studies converge in suggesting that crystallized abilities mediate the association between preschool fluid abilities and early school achievement. However, other studies failed to find such a mediation when considering a more extended time period. For instance, one study found that growth in crystallized abilities across ages 5 to 24 was independent of previous fluid abilities [18]. The mediation hypothesis could thus be limited to the early school years; as the contribution of preschool crystallized abilities to later school achievement decreases over time, the mediational role of crystallized abilities may also diminish over the primary school years.

This time-limited mediation hypothesis has never been documented within a developmentally informed design covering the whole primary school period. Furthermore, such a putative process should be investigated longitudinally while considering the well-documented stability of individual differences in school achievement [12, 33]. For instance, in one study, the predictive contribution of fluid abilities (IQ, working memory) to later school achievement was reduced considerably, but remained significant when the stability of school achievement was taken-into-account [34]. Thus, school achievement autoregressive pathways should be considered to determine how and when early fluid and crystallized abilities contribute to later school achievement. Longitudinal samples with repeated achievement measures are key to answering this question.

The present study aimed to address the question of the predictive association between early cognitive abilities and later school achievement in two longitudinal samples with extended repeated measures of school achievement across the primary school years. The general goal was to assess the contribution of specific preschool fluids abilities (cognitive visual-spatial skills, cognitive flexibility, short-term memory) and crystallized abilities (knowledge of numbers, letters, colors and shapes) to school achievement across primary school. We hypothesized that these contributions would co-evolve within a mediational design and vary across primary school; specifically 1) that crystallized abilities would partly mediate the predictive association between early preschool fluid abilities and later achievement in the early school years, but not necessarily in the later years of primary school, 2) that school achievement would become highly stable by the end of primary school, and 3) that beyond early crystallized abilities, early

fluid abilities would still modestly contribute to school achievement in the later grades of primary school.

## Method

### Participants

Two distinct longitudinal samples were used in this study: The Quebec Newborn Twin Study (QNTS) and the Quebec Longitudinal Study of Child Development (QLSCD). The QLSCD was used to replicate and extend the results obtained within the QNTS, here considered as a convenient sample. The QLSCD allowed to confirm and broaden the initial findings in several ways: 1) by verifying the initial findings in an age-homogeneous, population-wide representative sample of children, 2) by examining earlier assessments of preschool fluid abilities (at 41 months of age, see below), and 3) by providing a Quebec-wide standardized assessment of school achievement at the end of primary school (i.e., 6th grade Quebec national exams). The first convenient sample, the QNTS, was a population-based longitudinal study of twins born between April 1995 and December 1998 in the Greater Montreal Area, Canada [35]. All parents living in the Greater Montreal Area were contacted either by phone or letter. A total of 989 families were contacted, and 662 families (1324 twins) agreed to be part of the study. The sample was mainly composed of European descent (84%) and the first language of the parents was mainly French or English (81.3%). The average household income was around CAN $54,000, which is slightly higher than the average income of families in the same region at that time. Most parents had graduated from high school, as only 12% of mothers and 9.8% of fathers did not complete high school. A total of 1072 twins (435 monozygotic, 633 dizygotic; 49.3% girls) were assessed in the laboratory at 63 months ($M$ age = 63.60 months, $SD$ = 3.14), just before school entry, on various relevant fluid and crystallized cognitive measures. School achievement was then assessed in grade 1 (645 twins), grade 3 (754 twins), grade 4 (779 twins) and grade 6 (627 twins).

The second sample, the QLSCD, is a prospective longitudinal study of children who were sampled to be representative of the population of Quebec [33]. In 1998, all singleton infants between 59 and 60 gestational weeks of age in the Province of Quebec were targeted, except for infants in the Far North region, Cree and Inuit regions, and those in Aboriginal reserves. A sample of 2223 infants and their families selected through a region-based stratified sampling initially participated in the study; 80% of these infants lived in intact two-parent families, 10.8% in stepfamilies, and 9.2% in single-parent families. The mean age of mothers was 28.8 years and the mean age of fathers was 31.9 years. Children were from Canadian origins (67.7%), European descent (48.6%), African or Haitian origins (20.5%) or other origins (20.5%; participants could declare more than one ethnic origin). The language most often spoken at home was mainly French or English (85.3%). Most parents had graduated from high school, as only 17.9% of mothers and 17.6% of fathers had no high school diploma. Children of the QLSCD were then quasi-yearly assessed longitudinally. The current study (1583 participants, 51.7% girls) considered extensive cognitive assessments at 41 months ($M$ age = 41.10 months, $SD$ = .53), and then in kindergarten ($M$ age = 73.8 months, $SD$ = 3.05), as well as school achievement ratings assessed in grade 1 (1296 participants), grade 2 (1277 participants), grade 4 (950 participants) and grade 6 (982 participants).

### Procedure

In the QNTS, twins were administered a battery of tests of fluid and crystallized abilities in a laboratory setting at 63 months, just before entering kindergarten. The order of testing was counterbalanced, except for the *Block Design* test which was administered first. Twins were

tested by two different interviewers, in two different rooms, to avoid inflated twin similarity. Teachers' ratings of school achievement were then collected at the end of each school year, which allowed the teachers to be well acquainted with how their students were doing academically. In the QLSCD, children were administered a battery of tests of several fluid and crystallized ability measures at 41 months of age (fluid abilities only), the end of kindergarten (both fluid and crystallized abilities at 73 months on average, 13 months later than the QNTS children), in the school setting or at home. The order of the tests was counterbalanced. As in the QNTS, teachers' ratings of school achievement were collected at the end of each school year (i.e., grades 1, 2, 4 and 6). In addition, administrative records of the grade 6 uniform provincial exams in reading, writing and mathematics were also available. Except for these grade 6 provincial exams, the same instruments were used in the two samples and administered in French or English.

### Ethical considerations

Parental written informed consent and child verbal assent were obtained from ethics review boards at Université Laval, Québec City, and at Université de Montréal and Ste-Justine Hospital, Montréal, for all data collections.

### Measures

**School achievement.**   In the spring of four different grade levels in primary school, school achievement was assessed by teachers (i.e., a different teacher at each school year) using a similar set of questions. The teachers rated each child's achievement in four school subjects: mathematics, writing, reading, and overall performance, relative to his or her classmates. Scores were on a 5-point Likert scale where 1 indicated *clearly under average*, 3 indicated *average*, and 5 indicated *clearly above average*. Given the high correlations between the ratings of the four school subjects (*rs* ranged from .69 to .98), these scores were averaged to yield a general school achievement score for a given year. Teacher ratings of school achievement are generally reliable and valid, as indicated by their substantial correlations with other measures of academic achievement scores, including standardized measures of performance. A meta-analysis estimated that teacher's assessments of student's school achievement is highly related to their actual test performance (average *r* = .62) [36]. A recent study reported that teacher assessments of achievement correlated strongly phenotypically (*r*~.70) and genetically (genetic correlation~.80) with school achievement; compared to standardized exam results, they were also as reliable and stable at every stage of the educational experience, but <u>more</u> predictive of later achievement [37]. In the present study, internal consistency of school achievement scores was excellent with Cronbach's alpha ranging from .94 to .95.

In the QLSCD, school achievement in grade 6 was also measured by national standardized exams at the end of elementary school. These uniform exams were made available to the Quebec Institute of Statistics (QIS) for the QLSCD participants. They were then systematically scored by trained research assistants from the QIS to control for potential teacher/school biases. Children were tested on reading, writing, and mathematics. The reading test consisted of 12 questions rated on a scale of 0 to 3. The writing test implied writing a short text that was evaluated on a scale of 1 (unsatisfactory) to 5 (very satisfactory) for 5 criteria (relevance of ideas, organization of the text, syntax and punctuation, vocabulary, spelling). The mathematics test entailed 9 problems aimed at measuring several mathematical skills: understanding the task, mobilizing concepts, explaining the solution, analyzing and making choice, applying and justifying the solution. All mathematical problems were scored on a scale of 0 to 100 according to the requested skills. All reading, writing and mathematics scores were standardized on a

scale from 0 to 100, and then averaged into a reliable general school achievement score (Cronbach's alpha = .79). There was a high and significant correlation between this resulting direct assessment of school achievement and teacher ratings of school achievement in grade 6 ($r$ = .65). Given this high convergence, these two measures were modeled to reflect a latent factor of general school achievement in the following path analyses.

**Crystallized abilities.** The Lollipop test consists of 52 questions divided into four subtests each assessing a component of preschool crystallized abilities: a) Identification of Colors and Shapes; b) Picture Description and Spatial Recognition; c) Identification of Numbers and Counting and d) Identification of Letters and Writing [38]. An overall score can be obtained by combining the scores of the four subtests. The scores for subtests 1 to 3 ranged from 0 to 17, scores for subtest 4 ranged from 0 to 18, and overall scores ranged from 0 to 69. A French and an English version were used according to the language of the child. The overall score was used. Cronbach alpha for this total score was $\alpha$ = .73 in the twin study and $\alpha$ = .55 in the singleton sample. The modest Cronbach alpha in the older singleton sample (i.e., by 13 months on average) was caused by ceiling effects in some subscales of the test. This measure has been shown to successfully predict early school achievement [14, 17].

The *Peabody Picture Vocabulary Test–Third Edition* is a standardized measure of language frequently used in research and clinical settings to assess receptive vocabulary (PPVT–III) [39]. The version used in this study contained 170 cards each depicting four different actions, objects or emotions. Children were asked to point at the image that corresponds to the word said by the tester. One point was awarded for each correct answer. The test ends after six errors out of eight trials. This measure has been widely used with success in predicting different outcomes like school achievement and school readiness [17, 40]. The PPVT-III has good internal consistency, good concurrent validity, good test-retest reliability and has been validated with both French and English-speaking Canadian youths [41, 42].

Number knowledge was assessed using a shortened version of the *Number Knowledge Test* (NKT) [40]. The NKT measured the understanding of the base 10 system of whole numbers. The items of the NKT are divided into two levels of increasing difficulty. The first level has five items, and the second level has 13 items. The child had to answer successfully at least three items at the first level to proceed to the second level. The test ended after three consecutive failures. The total score corresponded to the number of successful responses. The NKT has high internal consistency, good test-retest reliability, good convergent validity with other measures of mathematics ability and has been shown to successfully predict school achievement in mathematics [12, 43].

**Fluid abilities.** Cognitive visual-spatial skills were assessed through the Block Design subtest of the *Wechsler Preschool and Primary Scale of Intelligence—Revised* (WPPSI-R) [42]. Considered to be the most representative non-verbal component of general cognitive abilities, the Block Design subtest is composed of 14 models that the child must reproduce using bicolored blocks. The test ends after 3 consecutive failures and bonus points can be gained as a function of time. Raw scores vary from 0 to 42 and are standardized according to the exact age of the child. This performance Scale subtest has good internal consistency, test-retest reliability and it is highly correlated with the total score of the WPPSI-R ($r$ = .62) [43]. Cognitive visual-spatial skills, assessed by the Block Design, have been shown to be highly predictive of different outcomes, including number knowledge and school achievement [12, 17]. The Block Design was administered at 63 months in QNTS, and twice, at 41 months and 73 months of age in QLSCD.

Short-term memory was assessed with the *Visually Cued Recall Task* (VCR) [44]. During the VCR, children had to memorize and recall items that were presented to them. In the first trial, the tester pointed two items among a series of items and asked the child to recall them. For each new trial, the number of items to be recalled in a new set of items was increased by

one. The test ended after two consecutive errors or if the child reached the maximum number of items to be recalled (10 items). The trials were presented in the same order for all participants and the items to be recalled were the same for all participants. The VCR score corresponded to the number of items recalled in the last trial completed by the participants (i.e., trial 10 or the test immediately preceding two consecutive incorrect tests). The VCR has also been linked to other measures of short-term or working memory [45], and shown to predict number knowledge trajectories [12]. As for the Block Design, the VCR was administered at 63 months in QNTS, and twice, at 41 months and 73 months of age in QLSCD.

Cognitive flexibility was assessed only in the twin sample with the *Dimensional Change Card Sort* (DCCS) [46], a widely used measure of executive functions. During the DCCS, children are shown two cards (e.g., a blue rabbit and a red boat) and must sort the following cards according to one of the dimensions (color or form). After a while, the rule changes and the children must sort out the cards according to the other dimension [46]. Performance was measured by the correct number of responses following the rule change. DCCS has good test-retest reliability and good convergence validity [47]. A reflect and logarithm transformation was used to reduce negative skewness. The DCCS was administered at 63 months in QNTS, and only at 73 months in QLSCD.

**Control variables.** Maternal education, household income and child sex were used as control variables in both cohorts. In the QNTS, maternal education (highest diploma obtained) and household income were obtained during a home interview with the mothers, when the twins were 6 months old. Maternal education was aggregated into a 4-point scale (from 1 to 4): from "high school not completed" to "university diploma obtained", and household income was aggregated into a scale from 1 to 11: from "less than 5000$" to "more than 80 000$". In the QLSCD, maternal education and household income were collected through questions from the National Longitudinal Survey of Children and Youth, when the child was 5 months old. Maternal education was measured as their highest diploma obtained, as revealed by a 4-point self-report scale ranging from 1 (no high school diploma) to 4 (university degree). Household income was assessed on a scale from 1 to 10: from "less than 10 000$" to "more than 80 000$". The child's sex (0 = boy, 1 = girl) was also measured and used as a control variable in both samples.

## Analyses

All analyses were conducted using the statistical software Mplus (version 8) and missing data were considered using the Full information maximum likelihood (FIML) estimation. In the QNTS, the percentage of missing data ranged from 19.03% to 41.51%, while in the QLSCD, the percentage of missing data varied from 6.76% to 40.11%. Missingness largely varied across measures (15 measures in QLSCD, 13 measures in QNTS), participants, time, and cohort. Data were mostly missing for school achievement measures. S1 Table documents the relation between missingness and the various measures by comparing three groups of participants: 1) participants without any missing data; 2) participants with only 1 and 2 missing data; 3) participants with more than 2 missing data. S1 Table indicates that missingness was associated with lower family income, visual-spatial skills, vocabulary, school achievement, and being male in both cohorts, as well as with lower mother education, short term memory at 41 months, and Lollipop scores in QLSCD.

The general plan of analysis was as follows. First, we examined the correlations between all variables. Second, through path analysis, we tested the likelihood of the hypothesized model regarding the prediction of school achievement at different grades. We used the comparative fit index (CFI), the Tucker-Lewis index (TLI), and the small Root Mean Square Error of Approximation (RMSEA) to test the fit of the model. Direct and indirect paths of the

contribution of fluid abilities to school achievement through the mediation of crystallized abilities were examined. Also, stability of school achievement was estimated by adding autoregressive paths between all previous and subsequent school grades achievement measures. Control variables (maternal education, household income and sex) were added in the model to control for their association with cognitive abilities (fluid and crystallized) and school achievement measures. In the QNTS, we used a robust estimator to take into account the non-independence of the twin measures [48]. Finally, we estimated the various hypothesized pathways by bootstrapping (1000 times) [49]. Bootstrapping calculates probable estimates of the direct and indirect contributions by generating multiple empirical representations of the sampling distribution [49]. These main analyses were followed by a series of specific sub-analyses to verify the robustness of the resulting model.

## Results

Table 1 present sample sizes, means and standard deviations for school achievement scores, crystallized abilities, fluid abilities and control variables for both the QNTS and the QLSCD. Crystallized abilities were found to be generally higher in the QLSCD since they were assessed at a later age (73 months) than in the QNTS (63 months). However, fluid abilities and school achievement had similar means across samples.

### Correlations

Table 2 presents the Pearson correlations between school achievement, crystallized abilities, fluid abilities and the control variables in both samples. Correlations were significant, although

**Table 1. Descriptives of the variables in the two samples.**

| | QNTS | | | QLSCD | | |
|---|---|---|---|---|---|---|
| | **M** | **SD** | **N** | **M** | **SD** | **N** |
| *Fluid abilities* | | | | | | |
| Block Design 41M | - | - | - | 6.29 | 3.82 | 1476 |
| Block Design 63/73M | 19.78 | 6.59 | 868 | 19.99 | 9.88 | 1144 |
| DCCS | 24.17 | 7.30 | 791 | - | - | - |
| VCR 41M | - | - | - | 3.25 | 2.20 | 1429 |
| VCR 63/73M | 4.89 | 2.21 | 843 | 5.92 | 2.36 | 1144 |
| *Crystallized abilities* | | | | | | |
| Lollipop | 42.67 | 13.05 | 865 | 57.85 | 7.06 | 1141 |
| PPVT | 54.64 | 18.79 | 733 | 80.66 | 17.15 | 1109 |
| NKT | 7.82 | 4.16 | 864 | 13.33 | 3.26 | 1132 |
| *School Achievement* | | | | | | |
| SA1 | 3.27 | 1.07 | 645 | 3.69 | 1.08 | 1296 |
| SA2 | - | - | - | 3.67 | 1.06 | 1277 |
| SA3 | 3.11 | 1.06 | 754 | | | |
| SA4 | 3.04 | 1.09 | 779 | 3.59 | 1.05 | 950 |
| SA6 | 3.26 | 1.10 | 627 | 3.56 | 1.06 | 982 |
| National Exams | - | - | - | 72.43 | 13.66 | 948 |
| *Control Variables* | | | | | | |
| Mother Education | 2.64 | 1.09 | 991 | 2.69 | 1.06 | 1582 |
| Family Income | 6.97 | 2.20 | 959 | 5.93 | 2.24 | 1564 |

*Note*. Block Design and VCR were measured at 63 months in QNTS and at 41 and 73 months in QLSCD; DCCS = Dimensional Change Card Sort; VCR = Visually Cued Recall Task; PPVT = Peabody Picture Vocabulary Test–Third Edition; NKT = Number Knowledge Test; SA = School Achievement.

**Table 2. Correlations between all variables in the QNTS (under the diagonal) and in the QLSCD (above the diagonal).**

| Variables | BD 41 | BD 63/73 | DCCS | VCR 41 | VCR 63/73 | Lollipop | PPVT | NKT | SA1 | SA2/SA3 | SA4 | SA6 | SA6 (exams) | Sex | Education | Income |
|---|---|---|---|---|---|---|---|---|---|---|---|---|---|---|---|---|
| *Fluid abilities* | | | | | | | | | | | | | | | | |
| BD 41 | - | .36* | n/a | .35* | .11* | .26* | .26* | .33* | .31* | .32* | .30* | .28* | .27* | .11* | .19* | .20* |
| BD 63/73 | n/a | - | n/a | .19* | .11* | .32* | .31* | .37* | .38* | .39* | .36* | .32* | .29* | -.03 | .23* | .21* |
| DCCS | n/a | .21* | - | n/a | n/a | n/a | n/a | n/a | n/a | n/a | n/a | n/a | n/a | n/a | n/a | n/a |
| VCR 41 | n/a | n/a | n/a | - | .12* | .23* | .28* | .32* | .29* | .29* | .23* | .21* | .28* | .21* | .23* | .20* |
| VCR 63/73 | n/a | .22* | .16* | n/a | - | .14* | .13* | .09* | .16* | .16* | .10* | .17* | .03 | .08* | .03 | .02 |
| *Crystallized abilities* | | | | | | | | | | | | | | | | |
| Lollipop | n/a | .49* | .25* | n/a | .27* | - | .41* | .51* | .58* | .53* | .45* | .42* | .31* | .18* | .23* | .25* |
| PPVT | n/a | .36* | .24* | n/a | .30* | .54* | - | .42* | .43* | .41* | .39* | .34* | .44* | .01 | .24* | .22* |
| NKT | n/a | .41* | .23* | n/a | .23* | .67* | .48* | - | .52* | .51* | .47* | .41* | .40* | .03 | .25* | .27* |
| *School Achievement* | | | | | | | | | | | | | | | | |
| SA1 | n/a | .43* | .14* | n/a | 26* | .58* | .42* | .51* | - | .78* | .70* | .62* | .55* | .11* | .30* | .29* |
| SA2/SA3 | n/a | .41* | .17* | n/a | .19* | .51* | .35* | .46* | .70* | - | .73* | .70* | .57* | .12* | .27* | .24* |
| SA4 | n/a | .49* | .17* | n/a | .25* | .54* | .41* | .46* | .69* | .78* | - | .71* | .73* | .14* | .26* | .23* |
| SA6 | n/a | .44* | .21* | n/a | .29* | .53* | .43* | .44* | .70* | .71* | .75* | - | .68* | .18* | .29* | .25* |
| SA6 (exams) | n/a | n/a | n/a | n/a | n/a | n/a | n/a | n/a | n/a | n/a | n/a | n/a | - | .20* | .32* | .35* |
| *Control Variable* | | | | | | | | | | | | | | | | |
| Sex | n/a | -.06 | -.03 | n/a | .12* | .06 | -.05 | -.03 | .07 | .11* | .13* | .15* | n/a | - | -.00 | .04 |
| Mother Education | n/a | .20* | .10* | n/a | .12* | .28* | .34* | .20* | .31* | .20* | .28* | .32* | n/a | -.01 | - | .53* |
| Family Income | n/a | .22* | .09* | n/a | .15* | .25* | .33* | .23* | .20* | .14* | .22* | .27* | n/a | -.02 | .47* | - |

*Note*. Block Design and VCR were measured at 63 months in QNTS and at 41 and 73 months in QLSCD; Grade 2 in QLSCD and grade 3 in QNTS; BD = Block Design; DCCS = Dimensional Change Card Sort; VCR = Visually Cued Recall Task; PPVT = Peabody Picture Vocabulary Test–Third Edition; NKT = Number Knowledge Test; SA = School Achievement; n/a = not applicable;

* $p < .05$.

modest among fluid abilities in both the QNTS and the QLSCD. In both samples, the correlations were significant and substantial among the various crystallized abilities (*rs* range: .41 to .67). Accordingly, they were modeled to reflect a latent factor of crystallized abilities in the path analysis model. The correlations between the control variables and school achievement were positive, and in the expected direction (e.g., sex: in favor of girls), although modest in magnitude (range: .07 to .35). School achievement was found to be highly stable across grades (*rs* range: .62 to .78), and teacher ratings in grades 1, 2, 4, as highly predictive of national exams in grade 6 (*rs* range: .55 to .73), which speaks to high convergence of these assessments over time, and the importance of considering auto-regressive paths in the predictive model. Finally, all early predictors were significantly associated with later school achievement, with the strongest correlations obtained for crystallized abilities (*rs* range: .31 to .58). It is interesting to note that both preschool fluid and crystallized abilities were significantly, and often substantially correlated to later (grades 4 and 6) school achievement.

## Prediction of school achievement

The proposed mediation model provided an adequate fit to the observed data in QNTS, as shown by a high comparative fit index (CFI = .98), a high Tucker-Lewis index (TLI = .95), as well as a small Root Mean Square Error of Approximation (RMSEA = .05, 90% CI [.03, .06];

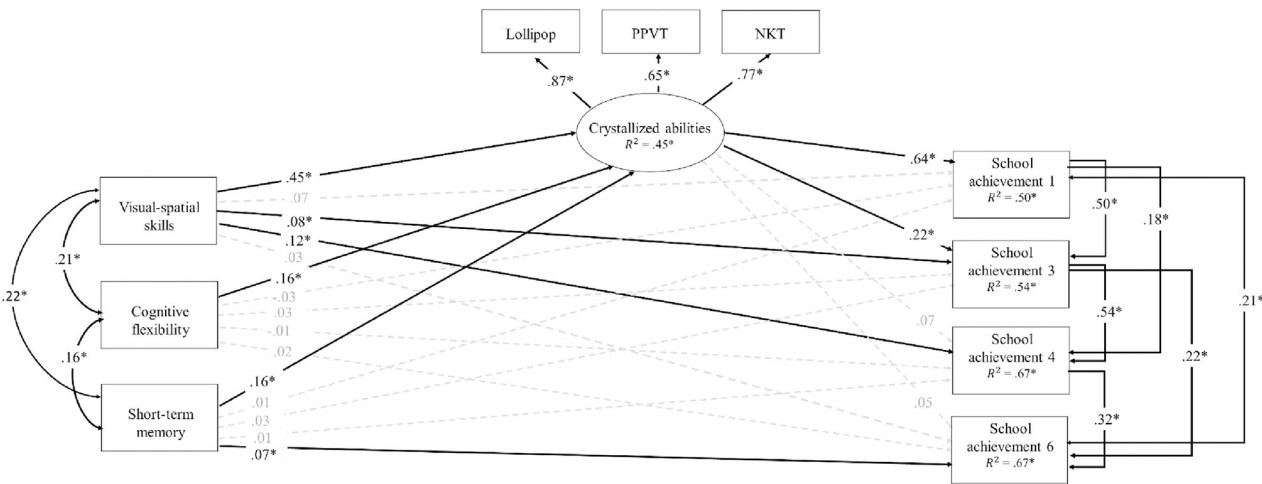

**Fig 1. Mediation model of the contributions of preschool fluid abilities and crystallized abilities to school achievement in the QNTS.** *Note.* $\chi^2(20)$ = 63.78 $p < .001$; RMSEA = .05, 90% CI [.03, .06]; CFI = .98, TLI = .95. Nonsignificant contributions are indicated with dashed lines. Contributions of control variables (mother's education, family income and sex) are not indicated to simplify the model. PPVT = Peabody Picture Vocabulary Scale, NKT = Number Knowledge Test.

[50]). Fig 1 shows the full model with the standardized path coefficients estimates for the QNTS.

The overall model, including the control variables, accounted for 50% of the variance in school achievement in grade 1, 54% in grade 3, and 67% in grades 4 and 6, with the latter grades including the autoregressive paths. Control variables made significant contributions to school achievement in grade 1 (maternal education only, $\beta = .11$, $p < .01$), in grade 3 (sex only, $\beta = .07$, $p < .02$) in grade 4 (maternal education only, $\beta = .06$, $p < .05$), and in grade 6 (maternal education, $\beta = .08$, $p < .02$, and sex, $\beta = .09$, $p < .01$). Over and above the contribution of these control variables, the model provided clear support for the mediation hypothesis. Indeed, the three measures of crystallized abilities clearly converged to create a reliable general latent crystallized abilities factor, which was significantly predicted ($R^2 = .45$) by the three fluid skills, and successfully predicted later school achievement. Specifically, early crystallized abilities (latent factor) were the strongest predictors of school achievement in grade 1 ($\beta = .64$, $p < .001$) and grade 3 ($\beta = .22$, $p < .001$). Over and above the substantial mediation, preschool fluid abilities also modestly predicted school achievement: cognitive visual-spatial skills predicted achievement in grade 3 ($\beta = .08$, $p < .04$) and grade 4 ($\beta = .12$, $p < .001$), and short-term memory predicted achievement in grade 6 ($\beta = .07$, $p < .01$). Interestingly, these modest direct contributions were significant beyond the contribution of early crystallized abilities and the strong auto-regressive (stability) paths characterizing school achievement in primary school. Finally, cognitive flexibility did not significantly contribute to any school achievement measures beyond early crystallized abilities.

Fig 2 shows the same full model for the QLSCD. Quite consistent with the QNTS results, the overall model explained 58% of the variance in school achievement in grade 1, 65% in grade 2, 60% in grade 4, and 80% in grade 6. In the QLSCD, Grade 6 school achievement was modeled as a latent factor derived from teacher ratings of school achievement and national exams. Control variables significantly contributed to school achievement in grade 1 (maternal education only, $\beta = .06$, $p < .05$), in grade 4 (sex only, $\beta = .04$ $p < .05$) and in grade 6 (sex, $\beta = .11$ $p < .001$; maternal education, $\beta = .07$, $p < .02$; household income $\beta = .08$, $p < .01$). Again, the model supported the mediation hypothesis, as the two fluid skills, already at 41 months ($\beta$

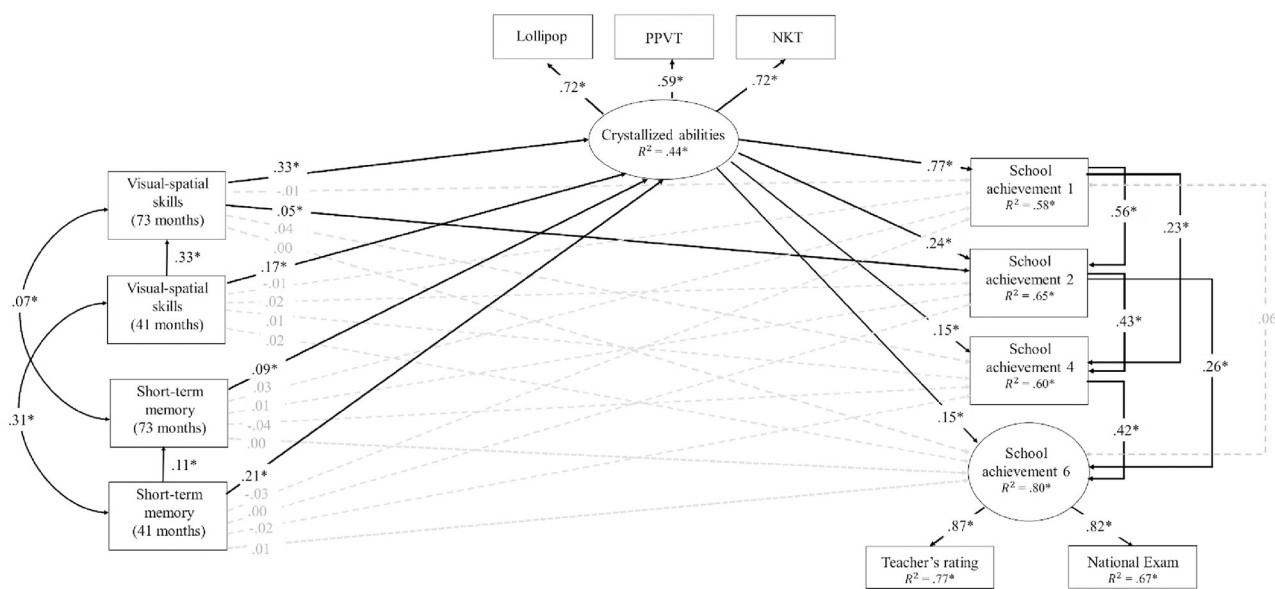

**Fig 2. Mediation model of the contributions fluid abilities and crystallized abilities to school achievement in the QLSCD.** *Note.* $\chi^2(37) = 173.69\ p < .001$; RMSEA = .05, 90% CI [.04, .06]; CFI = .98, TLI = .93. Nonsignificant contributions are indicated with dashed lines. Contributions of control variables (mother's education, family income and sex) are not indicated to simplify the model. PPVT = Peabody Picture Vocabulary Scale, NKT = Number Knowledge test.

= .17 and Beta = .21, $p < .001$, respectively) and then at 73 months ($\beta$ = .33 and Beta = .09, $p < .001$, respectively) each predicted crystallized abilities (latent factor; $R^2$ = .44), which also successfully predicted later school achievement in grade 1 ($\beta$ = .77, $p < .001$) and in grade 2 ($\beta$ = .24, $p < .001$). In contrast to QNTS, this prediction also extended to Grade 4 ($\beta$ = .15, $p < .03$) and to Grade 6 (as a latent factor; $\beta$ = .15, $p < .02$). Visual-spatial cognitive skills at 73 months also modestly predicted school achievement in grade 2 ($\beta$ = .05, $p < .03$), again beyond early crystallized abilities, and the strong auto-regressive (stability) paths characterizing school achievement in primary school. Short-term memory did not have a contribution to any school achievement measure beyond crystallized abilities.

Thus, the two models were quite supportive of the mediating role of crystallized abilities in the association between early fluid abilities and later school achievement. To estimate the magnitude of both direct and mediation pathways more precisely, we used a bootstrapping method. Table 3 shows these direct and indirect contributions and their confidence intervals in QNTS, while Table 4 shows these direct and indirect contributions and their confidence intervals in QLSCD, after a thousand bootstrapping iterations. Again, the results were similar in both samples. Over and above the direct contributions already described, all fluid abilities could be seen as having extended mediated contributions to school achievement across primary school, i.e., through crystallized abilities in grade 1, and then through crystallized abilities and previous achievement in the later grades (as posited in Figs 1 and 2). Specifically, for cognitive visual-spatial skills, the total mediated estimates varied from .27 to .30 in QNTS, and from .19 to .26 in QLSCD; for cognitive flexibility, they varied from .07 to .10 in QNTS (only); and for short-term memory, from .09 to .10 in QNTS, and from .06 to .17 in QLSCD. Thus, in both samples, crystallized abilities mediated the contribution of fluid abilities to school achievement in grade 1, which was then reverberated in later school achievement across primary school.

Finally, because crystallized abilities' contributions to later school achievement could not be reduced to previous fluid abilities, we also documented their unique direct and indirect

**Table 3. Direct and indirect effects of fluid abilities on school achievement through crystallized abilities and previous achievement in the QNTS sample.**

| Predictors | Outcome | Direct effect | | | Total indirect effect | |
|---|---|---|---|---|---|---|
| | | Coefficient | SE | P | Coefficient [95% CI] | SE |
| Cognitive visual-spatial skills | Achievement 1st | .07 | .04 | .13 | .29 [.22, .35] | .03 |
| | Achievement 3rd | .08 | .04 | .04 | .27 [.21, .34] | .03 |
| | Achievement 4th | .12 | .03 | .00 | .28 [.23, .35] | .03 |
| | Achievement 6th | .03 | .04 | .38 | .30 [.24, .37] | .03 |
| Cognitive flexibility | Achievement 1st | -.03 | .03 | .31 | .10 [.06, .15] | .02 |
| | Achievement 3rd | .03 | .03 | .35 | .07 [.03, .12] | .02 |
| | Achievement 4th | -.01 | .03 | .86 | .08 [.03, .13] | .03 |
| | Achievement 6th | .02 | .03 | .52 | .07 [.02, .12] | .03 |
| Short-term memory | Achievement 1st | .01 | .04 | .78 | .10 [.06, .15] | .02 |
| | Achievement 3rd | .03 | .03 | .33 | .09 [.05, .14] | .02 |
| | Achievement 4th | .02 | .03 | .56 | .10 [.05, .15] | .03 |
| | Achievement 6th | .07 | .03 | .01 | .10 [.05, .14] | .02 |

*Note*. The estimates are all standardized. SE = standard error; *p* = p-value; CI = confidence interval.

contributions to school achievement controlling for early fluid abilities (and other socio-demographic controls). Table 5 shows these direct (also see Figs 1 and 2) and indirect contributions in both QNTS and QLSCD, again after bootstrapping iterations. Thus, over and above the direct and indirect contributions of fluid abilities, crystallized abilities contributions to school achievement extended across primary school, but through previous achievement in the

**Table 4. Direct and indirect effects of fluid abilities on school achievement through crystallized abilities and previous achievement in the QLSCD sample.**

| Predictors | Outcome | Direct effect | | | Total indirect effect* | |
|---|---|---|---|---|---|---|
| | | Coefficient | SE | P | Coefficient [95% CI] | SE |
| Cognitive visual-spatial skills 41M | Achievement 1st | -.01 | .03 | .62 | .21 [.16, .26] | .03 |
| | Achievement 2nd | .02 | .02 | .35 | .20 [.15, .24] | .02 |
| | Achievement 4th | .01 | .02 | .73 | .19 [.15, .23] | .02 |
| | Achievement 6th | .02 | .02 | .33 | .19 [.15, .24] | .02 |
| Cognitive visual-spatial skills 73M | Achievement 1st | -.01 | .03 | .77 | .26 [.20, .32] | .03 |
| | Achievement 2nd | .05 | .03 | .03 | .22 [.17, .27] | .02 |
| | Achievement 4th | .04 | .03 | .18 | .22 [.18, .27] | .02 |
| | Achievement 6th | .00 | .03 | .92 | .25 [.20, .30] | .03 |
| Short-term memory 41M | Achievement 1st | -.03 | .03 | .27 | .17 [.12, .23] | .03 |
| | Achievement 2nd | .00 | .02 | .93 | .13 [.09, .18] | .02 |
| | Achievement 4th | -.02 | .02 | .46 | .12 [.08, .16] | .02 |
| | Achievement 6th | .01 | .02 | .71 | .12 [.08, .16] | .02 |
| Short-term memory 73M | Achievement 1st | .03 | .03 | .22 | .07 [.02, .11] | .02 |
| | Achievement 2nd | .01 | .02 | .55 | .08 [.04, .12] | .02 |
| | Achievement 4th | -.04 | .03 | .10 | .07 [.04, .11] | .02 |
| | Achievement 6th | .00 | .02 | 1.0 | .06 [.02, .10] | .02 |

*Note*.

* = For the ease of presentation, we combined the two paths to crystallized abilities (PS), that is the direct link through PS and the indirect link through skills at 73 months; SE = standard error; *p* = p-value; CI = confidence interval.

**Table 5. Direct and indirect effects of crystallized abilities to school achievement during primary school.**

| Samples | Outcome | Direct effect | | | Indirect effect | |
|---|---|---|---|---|---|---|
| | | Coefficient | SE | *P* | Coefficient [95% CI] | SE |
| QNTS | Achievement 1st | .64 | .05 | .00 | .00 [.00, .00] | .00 |
| | Achievement 3rd | .22 | .07 | .00 | .32 [.24, .40] | .04 |
| | Achievement 4th | .07 | .06 | .24 | .41 [.33, .50] | .04 |
| | Achievement 6th | .05 | .07 | .47 | .40 [.32, .51] | .05 |
| QLSCD | Achievement 1st | .77 | .04 | .00 | .00 [.00, .00] | .00 |
| | Achievement 2nd | .24 | .07 | .00 | .43 [.36, .51] | .04 |
| | Achievement 4th | .15 | .07 | .03 | .46 [.38, .56] | .05 |
| | Achievement 6th | .15 | .07 | .02 | .48 [.40, .57] | .04 |

*Note.* SE = standard error; *p* = p-value; CI = confidence interval.

previous grades (as posited in Figs 1 and 2). Specifically, the total mediated estimates varied from .32 to .41 in QNTS, and from .43 to .48 in QLSCD.

## Sensitivity analyses

**Examining the role of missing data.** Missing data was significant and largely distributed across the measures and participants over time. Missingness was non-random as indicated in the Supplementary material. To verify the robustness of the outcome model, we tested it under stricter rules of participant inclusion: one condition only involved participants having one or two missing values; another, with participants for which all measures were available (i.e., list-wise). As shown in S1 to S4 Figs, the model withstood the test very well; strong mediation was again found in all models, and most of the significant pathways were confirmed, except for those of low magnitude in the listwise models who were not significant due to diminished power.

**Examining the putative role of a latent fluid factor.** In the proposed model, we derived a latent factor for crystallized abilities, but not for fluid abilities due to their moderate intercorrelation. This means that measures of fluid abilities yielded more measurement error, which could affect their predictive validity. Accordingly, we forced the calculation of a latent fluid factor to evaluate its contribution in the mediational model. The resulting models are presented in S5 and S6 Figs. In both QNTS and QLSCD, the resulting latent fluid factor 1) was biased toward visual-spatial skills (factor loadings of .65 in QNTS, as well as .59 and .57 in QSLCD were all higher than all other loadings), 2) was highly predictive of crystallized abilities (latent score), 3) with the latter mediating the contribution of the former to school achievement in the early school grades, but not after grade 2. Over and above this strong association with "latent" crystallized abilities, this latent factor also predicted school achievement in grade 3 and grade 4. However, the predictive paths directly leading to school achievement were unstable, as shown by their large confidence interval after bootstrapping. The calculation of such a latent fluid factor also allowed to test an alternate, reverse model, where fluid abilities were posited to mediate crystallized abilities. S7 (QNTS) and S8 (QLSCD) Figs illustrate this reverse model. In QNTS, the resulting model was again consistent with the important role of crystallized abilities in predicting early school achievement, and with a delayed prediction of later school achievement by fluid abilities. Indeed, as the putative central mediator, fluid abilities 1) did not significantly predict grade 1 school achievement, 2) only marginally predicted grade 3 school achievement, but 3) significantly predicted grade 4 school achievement (but not grade 6 school achievement). Posited as the initial point in the model, crystallized abilities were substantially

predictive of fluid abilities (same as in the original model), but over and above this strong association, it also predicted school achievement in grades 1, but not in grades 3, 4 and 6. The case of QLSCD is clearer: although well predicted by crystallized abilities as the putative mediator, the fluid factor score did not predict school achievement at any time in grade school. In contrast, bypassing fluid abilities, crystallized abilities significantly predicted grade 1, and more marginally grade 2 school achievement, but not later. In other words, the reverse model was not only unlikely on theoretical ground, but also it did not provide a convincing picture of the possible mediating role of fluid abilities.

**Estimating the unique role of controls.** Finally, all models were tested controlling for mother education, family income and sex. To estimate the general unique contributions of these variables, we compared the variance accounted for in the model when these controls were applied versus not. In QNTS, the $R^2$ was .45 (with controls) versus .39 (without controls) for Crystallized Abilities (latent score; 6% difference accounted for by the controls), and .67 (with controls) versus .65 (without controls) for School Achievement in Grade 6 (latent score; 2% difference accounted for by the controls), with very marginal changes in the predictive paths. In QLSCD, the $R^2$ was .44 versus .38 for Crystallized Abilities (latent score; 6% difference accounted for by the controls), and .80 versus .77 for School Achievement in Grade 6 (latent score; 3% difference accounted for by the controls), again with very marginal changes in the predictive paths. Thus, even if these controls accounted for a significant part of the total variance in the models, the model still performed very well when these controls were applied.

## Discussion

The goal of the present study was to document the contributions of preschool fluid and crystallized abilities to later school achievement across primary school, and to examine the mediating role of crystallized abilities in this sequence of predictive associations. Through path analyses, and controlling for sex, maternal education and family income, crystallized abilities were indeed found to mediate the association between early fluid abilities and later school achievement in the early grades of school. These results were not only quite similar in two distinct samples, but also quite robust to the various handling of missing data. Furthermore, the postulated mediation was clearly more tailored to the data than the reverse sequence (where fluid abilities were posited as the mediator). All of this confirms the relevance of considering the coordinated and sequential contributions of preschool fluid and crystallized abilities to later school achievement. These findings extend previous research in several ways.

First, the present findings underlie the central role of preschool crystallized abilities in predicting later school achievement. Not only did crystallized abilities predict early school achievement over and above fluid abilities, as shown in previous studies [1], but they also accounted for the most part of the selected fluid abilities' contribution to school achievement. A few studies had previously found results pointing in that direction, but the present findings confirmed that the mediation role of preschool crystallized abilities was substantial for general school achievement, and not just for domain-specific achievement, such as for mathematics [30, 31]. These findings, although correlational, are consistent with the idea that children with higher fluid abilities are better prepared to acquire, before school entry, relevant crystallized abilities, such as knowledge of numbers and letters, and then more ready to achieve in school. The general view that the acquisition of crystallized abilities stems partly from the "investment" of fluid abilities in the acquisition of crystallized abilities is not new [23, 24, 51], but the present study was one of the first to longitudinally document this process from preschool to well into grade school. We discuss possible alternative explanations of the findings later in the following section.

Secondly, this extended and detailed follow-up also allowed to uniquely qualify the longitudinal reach of preschool crystallized abilities, and thus of the mediation process. Indeed, the present study revealed that crystallized abilities did not add (QNTS) or added modestly (QLSCD) to the prediction of school achievement beyond grade 3. This time-limited contribution suggests that by grade 4, most, but not all benefits of early preschool crystallized abilities have been harvested and consolidated by children within the learning system.

Thirdly, the other important piece of evidence from the present study stems from the strong auto-regressive paths of school achievement. Already from the start (i.e., school entry), school achievement was highly stable, and by grade 4 and continuing in grade 6, the main predictors of school achievement were by far prior school achievement. It is likely that the school curriculum had moved children toward more sophisticated learning issues (e.g., knowledge of fractions or grammar rules) for which the early crystallized abilities were less directly relevant. However, the diminished unique contributions of early crystallized abilities to teachers' rating of school achievement beyond grade 3 does not mean that crystallized abilities are no longer relevant, but rather that they have become embedded in school achievement. Indeed, although they were not directly measured here, skills and knowledge acquired in school progressively becomes new, specific crystallized abilities which are reflected in children's overall school achievement. The main point here is that children with strong crystallized abilities enter school with a head start, and then tend to keep this advantage over time. Furthermore, in the QLSCD the additive contribution of early crystallized to school achievement could still be seen in the later grades, thus reflecting the long-term developmental reach of crystallized abilities.

Finally, despite the strong auto-regressive paths of school achievement and mediational role of crystallized abilities, there was some evidence of very modest contributions of fluid abilities beyond grade 3, specifically of visuo-spatial cognitive skills (grade 4) and short-term memory (grade 6), but in QNTS only. These findings are consistent with previous research showing the predictive association between fluid abilities and school achievement [10, 19], but expands it by showing that most of the extended reach of fluid abilities are indirect and mediated by crystallized abilities. The small additional predictive value of these fluid abilities may have to do with the increased complexity of reasoning strategies requested for school achievement in late primary school [52]. It remains to be seen whether these contributions are maintained or perhaps augmented after primary school. The lack of an association between cognitive flexibility and school achievement is consistent with other studies showing no association when controlling for crystallized abilities [53], and again suggests that this central executive function plays out developmentally through the acquisition of crystallized abilities.

## Implications for preventive intervention

Given the strong stability of school achievement, children who start school ahead in terms of crystallized abilities are more likely to perform well academically in the early years of schooling, and possibly beyond. The present findings thus highlight the importance of early preventive interventions aimed at those abilities in preschool. For example, evaluations of early intervention programs, such as the Head Start program and its variants [54–56], have shown that it is possible to enhance fluid and crystallized abilities in preschool children, as well as their future school achievement [57]. Clearly, most early intervention programs that were proven successful have targeted crystallized abilities, and our findings suggest that crystallized abilities should indeed be the primary target of intervention.

According to Tricot and Sweller [58] we should distinguish domain-general information, a form of biologically primary knowledge naturally acquired without instruction (such as fluid abilities), from biologically secondary, domain-specific knowledge which can be taught,

learned and memorized to a point where it provides enough guidelines to achieve in various learning areas (such as knowledge about numbers and literacy) [59]. They maintain that the goal of any education enterprise should be to foster domain-specific knowledge rather than improving generic skills. The present findings are partly in sync with this view as they demonstrate that mastering crystallized abilities (i.e., relevant domain-specific knowledge) starts in preschool, plays out in the early years of school and then becomes assimilated into stable school achievement. This assimilation process could flow naturally as the increased knowledge and expertise provide "internal guidance" in further learning, which is then reflected in scholastic achievement [59]. From a different angle, some have raise the possible role of "third variables", such as stable personal and environmental factors to account for this stability in cognitive skills and school achievement [60] (see below for an extended discussion).

But at the same time, the present study also strongly suggests that learning these domain-specific knowledge and skills also depends on the initial (preschool) levels of fluid abilities. From an early prevention perspective and to the extent that they can be improved early on [53], fluid abilities should be considered as targets of early intervention. In light of Cattell's Investment Theory [23], our findings indicate that two aspects should be considered in this endeavor: 1) the capacity for preschool and school-based interventions to enhance fluid abilities, and 2) the provision of learning opportunities (i.e., a stimulating environment) so that these fluid abilities can be "invested" in the mastering of relevant crystallized abilities. Accordingly, future studies should investigate the extent to which the predictive association between fluid and crystallized abilities depends on (i.e., is moderated by) learning opportunities and contexts (e.g., the use of quality childcare services); in other words, they should document to what extent individual differences in the early, baseline levels of basic fluid cognitive abilities interact with these early learning contexts to shape crystallized abilities.

Still, there is some divergence when contrasting correlational-longitudinal work to true experimental evidence. Whereas the former provides significant developmental continuity, there are persisting doubts about the long-term impact of these early interventions [61]. Indeed, the jury is still out regarding the extent to which early intervention can foster fluid abilities, and whether these gains can then be translated into increased school readiness and achievement. There is some experimental evidence that fluid abilities, especially fluid reasoning, working memory, and executive functions can be successfully trained [53, 62]. However, still debated is whether these new and early acquired skills can extend their reach to later functionally related behavior and learning, such as preschool crystallized abilities and school achievement (i.e., far-transfer effects) [63]. A recent meta-analysis revealed that reasoning training improves general reasoning ability, but with mixed results regarding a possible transfer to academic performance (far transfer effects) [64]. Most of these reviewed studies were conducted among adolescent and adults, with only one randomized controlled study of preschool-age children showing that reasoning training improved fluid intelligence (inductive, deductive, and analogical reasoning abilities), but not working memory [65].

Early interventions aimed at improving executive functions among young children are more frequent, but their impact is also limited. In a recent meta-analysis, Kassai and colleagues [63] provided evidence for the effectiveness of training executive functions, but not for the far-transfer of these abilities in related areas. Specifically for young preschool children, Scionti and colleagues [66] also reported an overall significant positive effect of cognitive training on executive functions, this time with some generalization across executive functions, but not to other psychological domains, such as learning prerequisites and behavioral aspects. Thus, although extant meta-analyses show that training executive function is possible, the transfer appears limited to tasks tapping the trained abilities (see also [67] for similar conclusions regarding school-based interventions).

This uncoupling of findings between longitudinal and intervention work calls for further inquiry into possible missing links. It has been suggested that some stable personal and environmental factors (e.g., SES) could account for the higher continuity in cognitive and school achievement found in longitudinal non-experimental research [60, 68, 69]. To this point, the mediation model accounted for an impressive portion of the variance over and above sex and SES variables. The additional sensitivity analyses indicated that these controls only played a moderate role in the prediction model, and interestingly, more so when predicting preschool crystallized abilities than school achievement in 6th grade. In other words, the resulting model could not be accounted for by these persistent personal (sex) and socio-economic characteristics.

Nevertheless, it is still possible that unmeasured personality and genetic factors may be at play [60, 70]. For instance, previous twin studies have shown that fluid abilities are more likely to be accounted for by genetic sources of influences than crystallized abilities who tend to be more closely associated with environmental sources of influence [14, 22, 71]. To the extent that crystallized abilities are more likely than fluid abilities to develop through the nurturing experiences, including family-wide experiences, we could expect children who initially lagged in early crystallized abilities (e.g., early numeracy and literacy), but with good fluid skills, to catch up when they are exposed to the school curriculum. This could result in fluid abilities playing a more persistent role than early crystallized abilities in later school achievement. Future longitudinal studies extending beyond primary school should investigate this possibility. It is interesting to note that when modeled as as a latent variable in supplementary analyses, fluid abilities were as predictive of early school achievement as individual fluid measures. This is consistent with Nguyen, Duncan, and Bailey [72] who found the prediction between executive functions and math achievement to be mostly accounted for a general latent factor of executive function. Such a central role of a higher-order latent score in prediction (versus specific components) is compatible with a possible genetic propensity underlying and accounting for the developmental continuity between fluid cognitive abilities and some core components of crystallized abilities and school achievement. But this is an empirical question for future research.

## Limitations

The present study should be interpreted in the context of its limitations. First, despite the plausible nature of the proposed sequence of associations, fluid abilities and crystallized abilities were assessed at the same time in QNTS, thus, limiting the formal test of the posited sequence of association. However, in QLSCD the fact that fluid abilities were assessed much earlier and significantly predicted crystallized abilities two years later is evidence in favor of the proposed model. Furthermore, the reverse model, the one in which fluid skills were posited as the mediators, did not provide convincing evidence. In any case, more precise longitudinal protocols, such as cross-lagged designs, or experimental (intervention) designs will be necessary to confirm the direction of this association. It is also important to keep in mind that the proposed mediational model does not rule out a possible degree of mutual influence between the two sets of cognitive abilities [73]. Second, crystallized abilities measures were more interrelated than fluid abilities measures. This rather strong cross-domain associations among crystallized abilities have been documented before and could reflect the combined role of differentially stable personal attributes and environmental developmental building blocks [1, 60]. Yet, this conceptual/functional divergence may have limited the capacity to center on a core fluid construct, and thus its predictive capacity. More generally, only a subset of fluid and crystallized abilities was considered in the present study. For instance, only non-verbal fluid abilities

were assessed. Verbal fluid abilities, such as verbal reasoning, may be differently associated with crystallized abilities and school achievement, and should be investigated in future research. Furthermore, we did not measure working memory per se, a significant predictor of school achievement [10]. However, the Block Design test, the measure of visual-spatial cognitive abilities and a good proxy of general IQ [43], was likely to capture a significant portion of nonverbal fluid abilities. Third, although there was minimal asymmetry in the data, the Lollipop and the NKT presented a slight asymmetry mainly due to some modest ceiling effects. We used MLR (robust) under Mplus, which is robust to non-normality, and then a log transformation on these scores. It is also likely that the calculation of a latent score of crystallized abilities minimized potential biases. Fourth, school achievement was assessed mostly through teacher ratings. However, recent research shows that teacher ratings are highly correlated with standardized assessment of school achievement [12, 36, 37], and the present study confirmed that these ratings were highly correlated and predictive of standardized national exams at the end of primary school. Fifth, the present study only covered school achievement in the primary school period, and the findings may thus be relevant only to that period. Results may differ in later grades (i.e., secondary school), as the contribution of early cognitive abilities to school achievement may be nonlinear [19]. Finally, like most longitudinal studies, the present study was affected by missing data. Detailed analyses revealed that data were not missing at random, with more vulnerable children more likely to have incomplete data. However, we used a full information maximum likelihood (FIML) method, and further controlled for socioeconomic status (maternal education, household income) and sex in the models to attenuate this limitation. More decisively, the fact that the mediation model did not change significantly as a function of missingness suggests that the use of FIML was appropriate. In the end, this differential missing data pattern may have constrained the variance estimates (i.e., leading to more conservative estimates).

## Conclusion

In the present study, we tested in two distinct samples a comprehensive mediation model examining the sequential role of preschool fluid and crystallized abilities in the pathways to school achievement. Preschool crystallized abilities were found to account for the association between fluid abilities and school achievement, but mostly in the early school years. School achievement was highly stable across primary school, with both crystallized and fluid abilities having modest unique contributions to school achievement in the later grades. These results lend qualified support for the importance of early interventions aimed at improving both preschool fluid and crystallized abilities.

## Supporting information

**S1 Table. Missing value patterns and variations in model variable scores.**
(DOCX)

**S1 Fig. Mediation model of the contributions of preschool fluid abilities and crystallized abilities to school achievement in the QNTS (listwise condition).**
(DOCX)

**S2 Fig. Mediation model of the contributions of preschool fluid abilities and crystallized abilities to school achievement in the QNTS (two missing value condition).**
(DOCX)

**S3 Fig. Mediation model of the contributions of preschool fluid abilities and crystallized abilities to school achievement in the QLSCD (listwise condition).**
(DOCX)

**S4 Fig. Mediation model of the contributions of preschool fluid abilities and crystallized abilities to school achievement in the QLSCD (two missing value condition).**
(DOCX)

**S5 Fig. Mediation model of the contributions of preschool fluid abilities (latent factor) and crystallized abilities to school achievement in the QNTS.**
(DOCX)

**S6 Fig. Mediation model of the contributions of preschool fluid abilities (latent factor) and crystallized abilities to school achievement in the QLSCD.**
(DOCX)

**S7 Fig. Mediation model of the contributions of preschool crystallized abilities and fluid abilities (latent factor) to school achievement in the QNTS (reverse model).**
(DOCX)

**S8 Fig. Mediation model of the contributions of preschool crystallized abilities and fluid abilities (latent factor) to school achievement in the QLSCD (reverse model).**
(DOCX)

## Acknowledgments

The authors are grateful to the children and parents and to the participating teachers and schools of the QNTS and QLSCD. Special thanks Mireille Jetté, Bertrand Perron, Nancy Illick, and Institut de la Statistique du Québec (ISQ) (for QLSCD), as well as to Jocelyn Malo and Marie-Élyse Bertrand (for QNTS) for coordinating the data collections over the years. Many thanks also to Hélène Paradis for data management and analysis.

## Author Contributions

**Conceptualization:** Philippe Carpentier, Célia Matte-Gagné, Amélie Petitclerc, Isabelle Ouellet-Morin, René Carbonneau, Jean Séguin, Sylvana Côté, Frank Vitaro, Richard E. Tremblay, Ginette Dionne, Michel Boivin.

**Formal analysis:** Philippe Carpentier, Sophie Aubé, Bei Feng.

**Funding acquisition:** Mara Brendgen, Simon Larose, Amélie Petitclerc, Isabelle Ouellet-Morin, Jean Séguin, Sylvana Côté, Frank Vitaro, Richard E. Tremblay, Ginette Dionne, Michel Boivin.

**Investigation:** Philippe Carpentier, Célia Matte-Gagné, Jean Séguin, Sylvana Côté, Frank Vitaro, Richard E. Tremblay, Ginette Dionne, Michel Boivin.

**Methodology:** Philippe Carpentier, Bei Feng, Michel Boivin.

**Project administration:** Sylvana Côté, Frank Vitaro, Richard E. Tremblay, Ginette Dionne, Michel Boivin.

**Resources:** Michel Boivin.

**Software:** Bei Feng.

**Supervision:** Ginette Dionne, Michel Boivin.

**Writing – original draft:** Philippe Carpentier.

**Writing – review & editing:** Geneviève Morneau-Vaillancourt, Sophie Aubé, Célia Matte-Gagné, Anne-Sophie Denault, Mara Brendgen, Simon Larose, Amélie Petitclerc, Isabelle Ouellet-Morin, René Carbonneau, Bei Feng, Jean Séguin, Sylvana Côté, Frank Vitaro, Richard E. Tremblay, Ginette Dionne, Michel Boivin.

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
