## [Decision Letter · Decision Letter 0]

14 Jan 2022

PONE-D-21-36376A sequential model of the contribution of preschool fluid and crystallized cognitive abilities to later school achievementPLOS ONE

Dear Dr. Morneau-Vaillancourt,

Thank you for submitting your manuscript to PLOS ONE. After careful consideration, we feel that it has merit but does not fully meet PLOS ONE’s publication criteria as it currently stands. Therefore, we invite you to submit a revised version of the manuscript that addresses the points raised during the review process.

The manuscript has been reviewed by two experts in the field, who are both enthusiastic about the topic and the overall quality of the paper. Reviewer 1 does have some suggestions for further improvement of the manuscript, particularly for including some additional/recent studies in the introduction as well as conducting some additional/alternative analyses to test the robustness of the findings.

I have some additional comments/suggestions from my own reading of the manuscript. Although I must say that I am not an expert in the field, so do let me know if I misunderstood parts of the manuscript in any way.

Did the authors consider potential confounds of (early measures of) IQ, such as socio-economic status or indices of input at home?The authors discuss stability of school achievement, but how about stability of IQ over time? Particularly at such a young age. Might time limited effects of IQ on school achievement also be understood in terms of instability of IQ for young children?I wondered about the measures of crystallized abilities. To me, some of these tasks seem a rather direct measure of skills that are later considered school achievement. Could the relation, or rather difference, between (early) school achievement and crystallized abilities be explained a bit more?It is mentioned (page 12) that there were ceiling effects for some of the measures. How might this have affected the results? Was anything done about this?In line with comments of reviewer 1, I noticed that there is a significant amount of missing data. Does the model presented hold when only cases with complete data (at least on the most relevant measures) are considered?Have the authors considered the distributions of the variables included in het model and the potential effect thereof on the results?How well can fluid and crystallized abilities actually be distinguish in young children? Whereas the crystallized abilities seem interrelated, as expected, correlations among the fluid measures are much lower. In fact, all the fluid measures correlated more strongly with the crystallized tasks than the other fluid measures. How might these correlations have affected the model/outcomes?Discussion: although a clear summary is provided of the findings, the authors might also say something about potential alternative explanations of the findings.Specifically concerning training, I wondered whether it is desirable, or even possible, to train such broad skills as crystallized and fluid abilities. See for example: Tricot, A., & Sweller, J. (2014). Domain-specific knowledge and why teaching generic skills does not work. *Educational Psychology Review, 26, *265-283. https://doi.org/10.1007/s10648-013-9243-1Minor point: could the constructs measured be mentioned in Table 1, rather than the specific instruments? For the control variables, some information is missing. Please submit your revised manuscript by Feb 28 2022 11:59PM. If you will need more time than this to complete your revisions, please reply to this message or contact the journal office at plosone@plos.org. Please include the following items when submitting your revised manuscript:A rebuttal letter that responds to each point raised by the academic editor and reviewer(s). You should upload this letter as a separate file labeled 'Response to Reviewers'.A marked-up copy of your manuscript that highlights changes made to the original version. You should upload this as a separate file labeled 'Revised Manuscript with Track Changes'.An unmarked version of your revised paper without tracked changes. You should upload this as a separate file labeled 'Manuscript'.

We look forward to receiving your revised manuscript.

Kind regards,

Madelon van den Boer

Academic Editor

PLOS ONE

Journal Requirements:

- https://psyarxiv.com/rvmt5/

In your revision ensure you cite all your sources (including your own works), and quote or rephrase any duplicated text outside the methods section. Further consideration is dependent on these concerns being addressed.

"Both studies were supported by grants from the Fonds de recherche du Québec – Société et Culture (FRQSC), Fonds de recherche du Québec – Santé (FRQS), the Social Science and Humanities Research Council of Canada (SSHRC), the Canadian Institutes for Health Research (CIHR), and Ste Justine Hospital’s Research Center. In addition, the QNTS was supported by funding from the National Health Research Development Program, Université Laval, and Université de Montreal. The QLSCD was supported by funding from the Gouvernement du Québec, the Lucie and Andre Chagnon Foundation, the Robert-Sauvé Research Institute of Health and Security at Work, and the Institut de la statistique du Québec." 

Reviewers' comments:

Reviewer's Responses to Questions

**Comments to the Author**

1. Is the manuscript technically sound, and do the data support the conclusions?

Reviewer #1: Yes

Reviewer #2: Yes

2. Has the statistical analysis been performed appropriately and rigorously? 

Reviewer #1: Yes

Reviewer #2: Yes

3. Have the authors made all data underlying the findings in their manuscript fully available?

Reviewer #1: No

Reviewer #2: No

4. Is the manuscript presented in an intelligible fashion and written in standard English?

Reviewer #1: Yes

Reviewer #2: Yes

5. Review Comments to the Author

Reviewer #1: The article covers an important topic of early cognitive development, and links its results to potential implications for education. One of the strengths of the study resides in its leveraging of longitudinal data and its attempt to tease out cognitive skills’ and domains’ structuring mechanisms. I enjoyed reading the paper and would recommend it for publication after some revisions and clarifications have been made. I address these following article’s sections, and then conclude with a more synthetic comment.

Abstract

The sentence “Both fluid and crystallized abilities were found to significantly predict school achievement, but only in the early school years” seems at odds with the paper’s conclusion (p. 10, Conclusion section): “…fluid abilities having a small unique contribution to school achievement in the later grades.”

Introduction

• The literature on skill development, school readiness, and later achievement has been somewhat nuanced. Authors’ proposed sequential model would gained in being framed within that more recent literature (see for example: Bailey, D. H., Duncan, G. J., Watts, T., Clements, D. H., & Sarama, J. (2018). Risky business: Correlation and causation in longitudinal studies of skill development. American Psychologist, 73(1), 81).

• Page 5: The brief mention of general intelligence deserves more elaboration. While the distinction between fluid and crystalized domains is adequately justified, how authors’ theory of fluid and crystalized abilities fit within their implicit theory of general cognitive abilities and interpretation and nature of human intelligence should be clarified. (see: Protzko, J., & Colom, R. (2021). A new beginning of intelligence research. Designing the playground. Intelligence, 87, 101559). Doing so would also strengthen the justification for the study’s mediational hypotheses (pg. 6-7).

Method

• Attrition in school achievement measures is relatively substantial and the authors acknowledged in the Discussion section that missingness might not be random. Documenting with a table the relations between attrition patterns and control variables would be informative.

• Furthermore, since data is missing on outcome variables, it was not entirely clear how helpful was FIML. Additional analyses could be conducted with listwise-deletion models, as well as with a sample having outcome values present across all time-points. Obtained results should be tabled and compared with FIML ones.

• While in the case of crystallized ability, measurement error is addressed by modelling a reflective factor, this was not done for fluid ability specific measures which are thus potentially noisier. Modeling the latter with a reflective factor would achieve three things: 1) alleviate this measurement error potential bias; 2) check for results’ robustness; and 3) bring extra evidence for domain-level effect (fluid ability).

• The possibility of reverse causation was alluded to by the authors in the discussion section. Obtained model fits and pathway estimates should be contrasted with those of alternative models taking fluid abilities (and fluid ability factor) as mediator for crystallized ability(ies) modeled as predictor.

• It would also be informative to know the decomposition of fluid abilities effect on school achievement in terms of indirect and direct effects’ relative proportions.

• Following Table 2, page-numbering is off.

Discussion

• As the authors address implications for prevention and intervention, it would be important to conduct further analyses controlling for a general ability factor (loading on fluid and crystallized ability indicators). This would also give the opportunity to contrast the stability of the effects of that general factor on school achievement with those of the predictor(s) and mediator(s).

Contrary to authors’ claim (p. 8, section Implications for prevention and intervention), whether or not early childhood programs (Head Start or others) have been successful in enhancing fluid and crystallized abilities or later school achievement is not at all settled. Since the authors aim to identifying the nature and timing of elements in school achievement causal chain (and by implication, what could be intervened upon and when), it seems crucial to ascertain effects’ precision and robustness of each measures of interest.

Reviewer #2: The manuscript reports on two major longitudinal studies each spanning 7-8 years and each comprising around 1000 participants. The conclusions drawn from the analyses are supported by the data.

The data have been appropriately and rigorously analyzed with pathanalytic modeling techniques implemented in the Mplus program.

Some restrictions on data access apply, but data may be obtained by filing a request to access from the Research Unit on Children’s Psychosocial Maladjustment Website.

The manuscript is well written in standard English and the presentation is structured in an appropriate manner.

This is a very interesting manuscript, which is likely to have major theoretical implications, which in turn can provide a basis for interventions to support development of school achievement. A considerable amount of research has been devoted to investigations of the so called “Investment theory” originally proposed by Raymond Cattell, but it has proven difficult to find strong evidence in support of the theory. However, the present study relies on two well-designed longitudinal studies to show that crystallized abilities mediate the relation between early fluid abilities and later school achievement. More concretely, the two studies show that children with higher fluid abilities are better able to acquire relevant crystallized abilities before school start, such as number or letter knowledge, which makes them prepared to achieve in school. This is the fundamental idea of the “investment” theory and the present study is one of the first to longitudinally document this process from preschool through primary school.

Another important finding is the strong auto-regressive paths of school achievement. School achievement was demonstrated to be highly stable, and half-way into primary school the main predictor of school achievement was prior school achievement. This strong stability of school achievement highlights the importance of early preventive interventions focused on both fluid and crystallized abilities in preschool and the first grades. While most successful early intervention programs have targeted crystallized abilities, the present findings indicate the importance both of the capacity for an early intervention to enhance fluid abilities, and of the provision of learning opportunities so that these fluid abilities can be “invested” in the mastering of relevant crystallized abilities.

The manuscript thus has the potential to contribute a stronger theoretical basis for research on the early phases of school achievement, but the powerful longitudinal design and technical skills demonstrated in analyses of data are further reasons for accepting the manuscript for publication.

6. PLOS authors have the option to publish the peer review history of their article (what does this mean?). If published, this will include your full peer review and any attached files.

Reviewer #1: No

Reviewer #2: No

---

## [Author Response · Author response to Decision Letter 0]

12 Jul 2022

Please see our response in attached letters.

---

## [Decision Letter · Decision Letter 1]

4 Aug 2022

PONE-D-21-36376R1A sequential model of the contribution of preschool fluid and crystallized cognitive abilities to later school achievementPLOS ONE

Dear Dr. Morneau-Vaillancourt,

Thank you for submitting your manuscript to PLOS ONE. After careful consideration, we feel that it has merit but does not fully meet PLOS ONE’s publication criteria as it currently stands. Therefore, we invite you to submit a final revised version of the manuscript that addresses the points raised during the review process.

 First of all, it is only fair to acknowledge the effort made by the authors to address the comments previously indicated by the reviewers. 

On this occasion, I am a new editor taking on this manuscript, but I am quite satisfied with the progress of the manuscript and with the current version. So we are almost at the point of full acceptance of the article. However, one last (simple) effort is still needed to achieve this. Please take heed of the latest clarifications from each reviewer, as well as this one from myself: Page 37: Given that at this key point in the discussion the authors are enthusiastic about school-based interventions aimed at improving fluency skills, it would be necessary to base this enthusiasm on at least a couple of recent studies that have achieved this desirable improvement. Using Sternberg's work alone (i.e., Reference #53) is not enough to encourage this perhaps unlikely possibility. 

We look forward to receiving your revised manuscript.

Kind regards,

Juan-Carlos Pérez-González, Ph.D.

Academic Editor

PLOS ONE

Journal Requirements:

Reviewers' comments:

Reviewer's Responses to Questions

**Comments to the Author**

1. If the authors have adequately addressed your comments raised in a previous round of review and you feel that this manuscript is now acceptable for publication, you may indicate that here to bypass the “Comments to the Author” section, enter your conflict of interest statement in the “Confidential to Editor” section, and submit your "Accept" recommendation.

Reviewer #1: All comments have been addressed

Reviewer #2: All comments have been addressed

2. Is the manuscript technically sound, and do the data support the conclusions?

Reviewer #1: Yes

Reviewer #2: Yes

3. Has the statistical analysis been performed appropriately and rigorously? 

Reviewer #1: Yes

Reviewer #2: Yes

4. Have the authors made all data underlying the findings in their manuscript fully available?

Reviewer #1: No

Reviewer #2: Yes

5. Is the manuscript presented in an intelligible fashion and written in standard English?

Reviewer #1: Yes

Reviewer #2: Yes

6. Review Comments to the Author

Reviewer #1: The authors were very responsive to the last round of comments and the paper is much improved. This has the potential to be very impactful on the field. I have a few remaining things to address in final revisions.

1) I had difficulty connecting Table 3 and 4 with Figure 1 and 2. Specifically, how were the indirect effects calculated with later school achievement outcomes (3rd, 4th, and 6th grade)?

For example: it seems that Table 3, for 3rd grade school achievement should show a Total Indirect Effect (TIE) of about .24 = (.45 x .64 x .50)+(.45 x .22); for 4th grade a TIE of about .16 = (.45 x .64 x .5 x .54)+(.45 x .64 x .18) + (.45 x .07); etc.

Perhaps, I am missing something. Greater transparency in how Table 3 and 4 estimates were obtained would be helpful.

2) The alternative model with fluid abilities modeled as a latent variable (Figure S5 & S6) yielded an indirect effect magnitude on School Achievement 1 on par with total indirect effects from individual fluid ability measures. This might be worth further elaboration in the discussion as it is still unclear whether interventions would be more effective targeting specific fluid abilities or more general influences contributing to that Gf factor, or both (e.g., Protzko, J. (2017), Effects of cognitive training on the structure of intelligence; and Nguyen, T., Duncan, R. J., & Bailey, D. H. (2019). Theoretical and methodological implications of associations between executive function and mathematics in early childhood).

3) Very minor comment: Arrows of the fluid abilities latent variable (Figure S6) point as if a formative factor was modeled (instead of an expected reflective factor).

Reviewer #2: I noticed a few annoying typos;

Line 310: Maternal education was aggregated into a 4-point scales (from 1 to 4)

Change to: Maternal education was aggregated into a 4-point scale (from 1 to 4)

Line 335: index (CFI), the Tucker-Lewis index (TLI), and the small root-mean-square error of approximation (RMSEA)

Change to: index (CFI), the Tucker-Lewis index (TLI), and the Root Mean Square Error of Approximation (RMSEA)

Line 374: as a small root-mean-square error of approximation (RMSEA

Change to: as a small Root Mean Square Error of Approximation (RMSEA

Line 403: <.001, respectfully) and then at 73 months (β = .33 and Beta = .09, p. <.001, respectfully)

Change to: <.001, respectively) and then at 73 months (β = .33 and Beta = .09, p. <.001, respectively)

Table 3. Direct and Indirect Effects of Fluids Abilities on School Achievement Through Crystallized Abilities and Previous Achievement in the QNTS Sample

Change to: Direct and Indirect Effects of Fluid Abilities on School Achievement Through Crystallized Abilities and Previous Achievement in the QNTS Sample

Table 4. Direct and Indirect Effects of Fluids Abilities on School Achievement Through Crystallized Abilities and Previous Achievement in the QLSCD Sample.

Change to: Table 4. Direct and Indirect Effects of Fluid Abilities on School Achievement Through Crystallized Abilities and Previous Achievement in the QLSCD Sample.

Line 638: and predictive of standardized national exams at the end of primary school. Fourth, the

Change to: and predictive of standardized national exams at the end of primary school. Fifth, the

7. PLOS authors have the option to publish the peer review history of their article (what does this mean?). If published, this will include your full peer review and any attached files.

Reviewer #1: No

Reviewer #2: No

---

## [Author Response · Author response to Decision Letter 1]

20 Sep 2022

Please see Response to Reviewers.

---

## [Editor Report · Decision Letter 2]

10 Oct 2022

A sequential model of the contribution of preschool fluid and crystallized cognitive abilities to later school achievement

PONE-D-21-36376R2

Dear Dr. Morneau-Vaillancourt,

We’re pleased to inform you that your manuscript has been judged scientifically suitable for publication and will be formally accepted for publication once it meets all outstanding technical requirements.

Kind regards,

Juan-Carlos Pérez-González, Ph.D.

Academic Editor

PLOS ONE

Additional Editor Comments (optional):

Great piece of work!

Glad to have edited the latest drafts.
---

## [Editor Report · Acceptance letter]

10 Nov 2022

PONE-D-21-36376R2 

A sequential model of the contribution of preschool fluid and crystallized cognitive abilities to later school achievement 

Dear Dr. Morneau-Vaillancourt:

I'm pleased to inform you that your manuscript has been deemed suitable for publication in PLOS ONE. Congratulations! Your manuscript is now with our production department. 

Kind regards, 

on behalf of

Dr. Juan-Carlos Pérez-González 

Academic Editor

PLOS ONE